# Pruning has a disparate impact on model accuracy

**Cuong Tran**
Department of Computer Science
Syracuse University
cutran@syr.edu

**Ferdinando Fioretto**
Department of Computer Science
Syracuse University
ffiorett@syr.edu

**Jung-Eun Kim**\*
Department of Computer Science
North Carolina State University
jung-eun.kim@ncsu.edu

**Rakshit Naidu**
Department of Computer Science
Carnegie Mellon University
rnemakal@andrew.cmu.edu

## Abstract

Network pruning is a widely-used compression technique that is able to significantly scale down overparameterized models with minimal loss of accuracy. This paper shows that pruning may create or exacerbate disparate impacts. The paper sheds light on the factors to cause such disparities, suggesting differences in gradient norms and distance to decision boundary across groups to be responsible for this critical issue. It analyzes these factors in detail, providing both theoretical and empirical support, and proposes a simple, yet effective, solution that mitigates the disparate impacts caused by pruning.

## 1 Introduction

As deep learning models evolve and become more powerful, they also become larger and more costly to store and execute. The trend hinders their deployment in resource-constrained platforms, such as embedded systems or edge devices, which require efficient models in time and space. To address this challenge, studies have developed a variety of techniques to prune the relatively insignificant or insensitive parameters from a neural network while ensuring competitive accuracy [1, 5, 7, 30, 31, 32, 40]. When a model needs to be developed to fit given and certain requirements in size and resource consumption, a pruned model which is derived from a large, rigorously-trained, and (often) over-parameterized model, is regarded as a de-facto standard. That is because it performs incomparably better than a same-size dense model which is trained from scratch, when the same amount of effort and resources are invested.

In spite of its strengths, pruning has been showed to induce or exacerbate disparate effects in the accuracy of the resulting reduced models [20, 19]. Intuitively, the removal of model weights affects the process in which the network separates different classes, which can have contrasting consequences for different groups of individuals. This paper further shows that the accuracy of the pruned models tends to increase (decrease) more in classes that had already high (low) accuracy in the original model, leading to a "the rich get richer" and "the poor get poorer" effect. This *Matthew* effect is illustrated in Figure 1. The figure shows the accuracy of a facial recognition task on different demographic groups for several pruning rates (indicating the percentage of parameters removed from the original models). Notice how the accuracy of the majority group (White) tends to increase while that of the minority groups tends to decrease as the pruning ratio increases.

Following these observations, we shed light on the factors to cause such disparities. The theoretical findings suggest the presence of two key factors responsible for why accuracy disparities arise in

---

\*This work was also conducted when Jung-Eun Kim was an assistant professor at Syracuse University.

36th Conference on Neural Information Processing Systems (NeurIPS 2022).

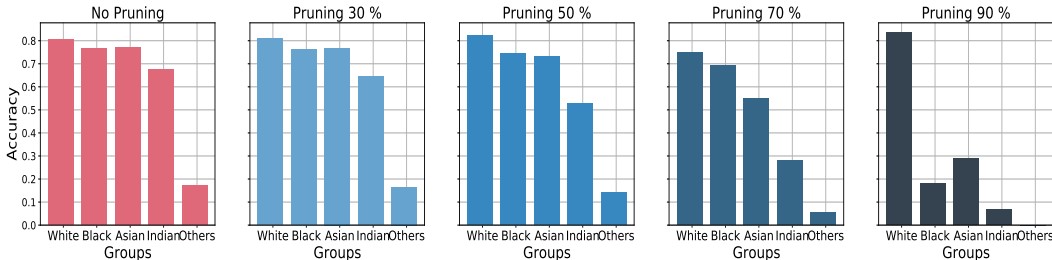

Figure 1: Accuracy of each demographic group in the UTK-Face dataset using Resnet18 [18], at the increasing of the pruning rate.

pruned models: **(1)** disparity in *gradient norms* across groups, and **(2)** disparity in *Hessian matrices* associated with the loss function computed using a group's data. Informally, the former carries information about the groups' local optimality, while the latter relates to model separability. We analyze these factors in detail, providing both theoretical and empirical support on a variety of settings, networks, and datasets. By recognizing these factors, we also develop a simple yet effective training technique that largely mitigates the disparate impacts caused by pruning. The method is based on an alteration of the loss function to include components that penalize disparity of the average gradient norms and distance to decision boundary across groups.

These findings are significant: *Pruning is a key enabler for neural network models in embedded systems with deployments in security cameras and sensors for autonomous devices for applications where fairness is an essential need. Without careful consideration of the fairness impact of these techniques, the resulting models can have profound effects on our society and economy.*

**Related work**

Fairness and network pruning have been long studied in isolation. The reader is referred to the related papers and surveys on fairness [4, 8, 11, 17, 24] and pruning [1, 5, 7, 30, 31, 32, 33, 40] for a review on these areas.

The recent interest in assessing societal values of machine learning models has seen an increase of studies at the intersection of different properties of a learning model and their effects on fairness. For example, Xu et al. [39] studies the setting of adversarial robustness and show that adversarial training introduces unfair outcomes in term of accuracy parity [42]. Zhu et al. [44] show that semisupervised settings can introduce unfair outcomes in the resulting accuracy of the learned models. Finally, several authors have also shown that private training can have unintended disparate impacts to the resulting models' outputs [3, 13, 34, 36, 43] and downstream decisions [29, 35].

Network compression has also been shown to have a profound impact towards the model fairness. For example, several works observed empirically that network compression may amplify unfairness in different learning tasks [27, 19, 20, 22]. Most of the focus has been on vision tasks and in identifying the set of *Pruning Identified Exemplars* (PIEs), the samples that are impacted most under the compression scheme and conclude that PIEs belongs to low frequency groups (those observed at the tail of the data distribution). Blakeney et al. [6] further investigate how bias could be evaluated and mitigated in pruned neural networks using knowledge distillation while Hosseini et al. [21] observed empirically that knowledge distillation processes may produce unfair student models. The impact of network compression towards fairness has also been assessed in natural language tasks. For example, Du et al. [10] and Xu et al. [37] empirically measure the robustness of compressed large language models, while Ahia et al. [2] look into how compression schemes affects data-limited regimes. Finally, Xu and Hu [38] investigate ways to improve fairness in generative language models by compressing them. We also note that, concurrently to this work, Good et al. [14] studied the relative distortion in recall for various classes. They show that pruning has a Matthews effect on the recall for various classes and propose an algorithm to attenuate such an effect.

This paper builds on this body of work and their important empirical observations and provides a step towards a deeper theoretical understanding of the fairness issues arising as a result of pruning. It derives conditions and studies the causes of unfairness in the context of pruning as well as it introduces mitigating guidelines.

## 2 Problem settings and goals

The paper considers datasets $D$ consisting of $n$ datapoints $(\boldsymbol{x}_i, a_i, y_i)$, with $i \in [n]$, drawn i.i.d. from an unknown distribution $\Pi$. Therein, $\boldsymbol{x}_i \in \mathcal{X}$ is a feature vector, $a_i \in \mathcal{A}$ with $\mathcal{A} = [m]$ (for some finite $m$) is a demographic group attribute, and $y_i \in \mathcal{Y}$ is a class label. For example, consider the case of a face recognition task. The training example feature $\boldsymbol{x}_i$ may describe a headshot of an individual, the protected attribute $a_i$ may describe the individual's gender or ethnicity, and $y_i$ represents the identity of the individual. The goal is to learn a predictor $f_{\boldsymbol{\theta}} : \mathcal{X} \rightarrow \mathcal{Y}$, where $\boldsymbol{\theta}$ is a $k$-dimensional real-valued vector of parameters that minimizes the empirical risk function:

$$\overset{\star}{\boldsymbol{\theta}} = \underset{\boldsymbol{\theta}}{\operatorname{argmin}} \, J(\boldsymbol{\theta}; D) = \frac{1}{n} \sum_{i=1}^{n} \ell(f_{\boldsymbol{\theta}}(\boldsymbol{x}_i), y_i), \tag{1}$$

where $\ell : \mathcal{Y} \times \mathcal{Y} \rightarrow \mathbb{R}_+$ is a non-negative *loss function* that measures the model quality.

We focus on analyzing properties arising when extracting a small model $f_{\bar{\boldsymbol{\theta}}}$ with $\bar{\boldsymbol{\theta}} \subset \overset{\star}{\boldsymbol{\theta}}$ of size $|\bar{\boldsymbol{\theta}}| = \bar{k} \ll k$. Model $f_{\bar{\boldsymbol{\theta}}}$ is constructed by pruning the least important values or filters from vector $\overset{\star}{\boldsymbol{\theta}}$ (i.e., those with smaller values in magnitude) according to a prescribed criterion, such as an $\ell_p$ norm [25, 31]. The paper focuses on understanding the fairness impacts (as defined next) arising when pruning general classifiers, such as neural networks.

**Fairness** The fairness analysis focuses on the notion of *excessive loss*, defined as the difference between the original and the pruned risk functions over some group $a \in \mathcal{A}$:

$$R(a) = J(\bar{\boldsymbol{\theta}}; D_a) - J(\overset{\star}{\boldsymbol{\theta}}; D_a), \tag{2}$$

where $D_a$ denotes the subset of the dataset $D$ containing samples $(\boldsymbol{x}_i, a_i, y_i)$ whose group membership $a_i = a$. Intuitively, the excessive loss represents the change in loss (and thus, in accuracy) that a given group experiences as a result of pruning. Fairness is measured in terms of the maximal *excessive loss difference*, also referred to as *fairness violation*:

$$\xi(D) = \max_{a, a' \in \mathcal{A}} |R(a) - R(a')|, \tag{3}$$

defining the largest excessive loss difference across all protected groups. (Pure) fairness is achieved when $\xi(D) = 0$, and thus a fair pruning method aims at minimizing the excessive loss difference.

The goal of this paper is to shed light on why fairness issues arise (i.e., $R(a) > 0$) as a result of pruning, why some groups suffer more than others (i.e., $R(a) > R(a')$), and what mitigation measures could be taken to minimize unfairness due to pruning.

We use the following notation: variables are denoted by calligraph symbols, vectors or matrices by bold symbols, and sets by uppercase symbols. Finally, $\| \cdot \|$ denotes the Euclidean norm and we use $f_{\boldsymbol{\theta}}(\boldsymbol{x})$ to refer to the model' *soft* outputs. All proofs are reported in Appendix A.

## 3 Fairness analysis in pruning: Roadmap

To gain insights on how pruning may introduce unfairness, we start with providing a useful upper bound for a group's excessive loss. Its goal is to isolate key aspects of model pruning that are responsible for the observed unfairness. The following discussion assumes the loss function $\ell(\cdot)$ to be at least twice differentiable, which is the case for common ML loss functions, such as mean squared error or cross entropy loss.

**Theorem 1.** *The excessive loss of a group $a \in \mathcal{A}$ is upper bounded by*[2]:

$$R(a) \leq \left\| \boldsymbol{g}_a^{\ell} \right\| \times \left\| \bar{\boldsymbol{\theta}} - \overset{\star}{\boldsymbol{\theta}} \right\| + \frac{1}{2} \lambda\left( \boldsymbol{H}_a^{\ell} \right) \times \left\| \bar{\boldsymbol{\theta}} - \overset{\star}{\boldsymbol{\theta}} \right\|^2 + O\left( \left\| \bar{\boldsymbol{\theta}} - \overset{\star}{\boldsymbol{\theta}} \right\|^3 \right), \tag{4}$$

*where $\boldsymbol{g}_a^{\ell} = \nabla_{\boldsymbol{\theta}} J(\overset{\star}{\boldsymbol{\theta}}; D_a)$ is the vector of gradients associated with the loss function $\ell$ evaluated at $\overset{\star}{\boldsymbol{\theta}}$ and computed using group data $D_a$, $\boldsymbol{H}_a^{\ell} = \nabla_{\boldsymbol{\theta}}^2 J(\overset{\star}{\boldsymbol{\theta}}; D_a)$ is the Hessian matrix of the loss function $\ell$, at the optimal parameters vector $\overset{\star}{\boldsymbol{\theta}}$, computed using the group data $D_a$ (henceforth simply referred to as group hessian), and $\lambda(\Sigma)$ is the maximum eigenvalue of a matrix $\Sigma$.*

---

[2]With a slight abuse of notation, the results refer to $\bar{\boldsymbol{\theta}}$ as the homonymous vector which is extended with $k - \bar{k}$ zeros.

The bound above follows from a second order Taylor expansion of the loss function, Cauchy-Schwarz inequality, and properties of the Rayleigh quotient.

Notice that, in addition to the difference in the original and pruned parameters vectors, two key terms appear in Equation (9): **(1)** The norms of the gradients $g_a^\ell$ and **(2)** the maximum eigenvalue of the Hessian matrix $H_a^\ell$ for a group $a$. Informally, the former is associated with the groups' local optimality while the latter relates to the ability of the model to separate the groups data. As we will show next these components represent the main sources of unfairness due to model pruning.

The following is an important corollary of Theorem 1. It shows that the larger the pruning, the larger will be the excessive loss for a given group.

**Corollary 1.** *Let $\bar{k}$ and $\bar{k}'$ be the size of parameter vectors $\bar{\theta}$ and $\bar{\theta}'$, respectively, resulting from pruning model $f_{\bar{\theta}}^\star$, where $\bar{k} < \bar{k}'$ (i.e., the former model prunes more weight than the latter one). Then, for any group $a \in \mathcal{A}$,*

$$\tilde{R}(a, \bar{\theta}) \geq \tilde{R}(a, \bar{\theta}'), \tag{5}$$

*where $\tilde{R}(a, \omega)$ is the excessive loss upper bound computed using pruned model parameters $\omega$ (Eq. (9)).*

The corollary above indicates that the excess risk for a group increases as the pruning regime increase. Building on this result, the paper illustrates next why unfairness can become more significant as the pruning regime increases.

The next sections analyze the effect of gradient norms and the Hessian to unfairness in the pruned models. The theoretical claims are supported and complemented by analytical results. These results use the UTKFace dataset [41] for a vision task whose goal is to classify ethnicity. The experiments use a ResNet-18 architecture and the pruning counterparts remove the *P%* parameters with the smallest absolute values for various *P*. All reported metrics are normalized and an average of 10 repetitions. While the theoretical analysis focuses on the notion of disparate impacts under the lens of excessive loss, the empirical results report differences in accuracy of the resulting models. The empirical results thus reflect the setting commonly adopted when measuring accuracy parity [42].

We report a glimpse of the empirical results, with the purpose of supporting the theoretical claims, and extended experiments, as well as additional descriptions of the datasets and settings, are reported in Appendix C.

## 4 Why disparity in groups' gradients causes unfairness?

This section analyzes the effect of gradients norms on the unfairness observed in the pruned models. In more detail, it shows that unbalanced datasets result in a model with large differences in gradient norms between groups (Proposition 1), it connects gradients norms for a group with the resulting model errors in such a group (Proposition 2), and connects these concepts with the excessive loss (Theorem 1) to show that unfairness in model pruning is largely controlled by the difference in gradient norms among groups.

**Gradient norms and group sizes.** The section first shows that imbalanced datasets lead a model to have imbalanced gradient norms across groups. The following result assumes that the training converges to a local minima.

**Proposition 1.** *Consider a dataset with two groups, a and b with $|D_a| \geq |D_b|$. Then $\left\|g_a^\ell\right\| \leq \left\|g_b^\ell\right\|$.*

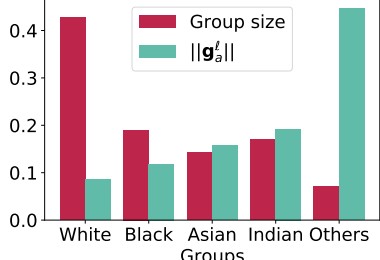

Figure 2: Group size vs. gradient norms.

That is, groups with more data samples will result in smaller gradients norms than groups with fewer data samples and vice-versa. Figure 2 illustrates Proposition 1. The plot shows the relation between groups sizes $|D_a|$ and their associated gradient norms $\|g_a^\ell\|$ on the UTK dataset and settings described above. Notice the strong trend between decreasing group sizes and increasing gradient norms for such groups. These theoretical considerations can be used to explain why underrepresented groups are often subject to larger performance impacts after network pruning [20]. These groups tend to exhibit large gradient norms at convergence, relative to other groups, thus, by Theorem 1, they are also subject to larger excessive losses due to pruning.

**Gradient norms and accuracy.** Next, the section shows a strong connection between the gradient norms of a group and its associated accuracy. The following assumes the models adopt a cross entropy loss (or mean squared error for regression tasks, as shown Appendix A).

**Proposition 2.** *For a given group $a \in \mathcal{A}$, gradient norms can be upper bounded as:*

$$\|g_a^\ell\| \in O\left( \sum_{(\boldsymbol{x},y)\in D_a} \underbrace{\|f_{\boldsymbol{\theta}}^\star(\boldsymbol{x}) - y\|}_{Error} \times \left\|\nabla_{\boldsymbol{\theta}} f_{\boldsymbol{\theta}}^\star(\boldsymbol{x})\right\| \right).$$

The above relates gradient norms with an error measure of the classifier to a target label multiplied by the gradient of the predictions. For example, in a classification task with cross entropy loss, $\ell(f_{\boldsymbol{\theta}}(\boldsymbol{x}), y) = -\sum_{z\in\mathcal{Y}} f_{\boldsymbol{\theta}}^z(\boldsymbol{x})\boldsymbol{y}^z$, where $f_{\boldsymbol{\theta}}^z(\boldsymbol{x})$ represents the $z$-th element of the output associated with the soft-max layer of model $f_{\boldsymbol{\theta}}$, and $\boldsymbol{y}$ is a one-hot encoding of the true label $y$, with $\boldsymbol{y}^z$ representing its $z$-th element, then,

$$\|g_a\| = \|\nabla_{\boldsymbol{\theta}} J(\boldsymbol{\theta}; D_a, )\| = \left\| {}^1\!/_{|D_a|} \sum_{(\boldsymbol{x},y)\in D_a} \nabla_f \ell(f_{\boldsymbol{\theta}}(\boldsymbol{x}), y) \times \nabla_{\boldsymbol{\theta}} f_{\boldsymbol{\theta}}(\boldsymbol{x}) \right\|$$

$$= \left\| {}^1\!/_{|D_a|} \sum_{(\boldsymbol{x},y)\in D_a} (f_{\boldsymbol{\theta}}(\boldsymbol{x}) - \boldsymbol{y}) \times \nabla_{\boldsymbol{\theta}} f_{\boldsymbol{\theta}}(\boldsymbol{x}) \right\|$$

$$\leq {}^1\!/_{|D_a|} \sum_{(\boldsymbol{x},y)\in D_a} \|f_{\boldsymbol{\theta}}(\boldsymbol{x}) - \boldsymbol{y}\| \times \|\nabla_{\boldsymbol{\theta}} f_{\boldsymbol{\theta}}(\boldsymbol{x})\| .$$

Figure 3: Accuracy vs. gradient norms.

A similar observation holds for mean square error loss, as illustrated in Appendix A. The observation above sheds light on the correlation between the prediction error of a group and its model gradients. This relation is emphasized in Figure 3, which illustrates that the gradient norm for a given group increases as its prediction accuracy decreases.

Proposition 2 allows us to link the gradient norms with the group accuracy of the resulting model, which, together with the result above will be useful to reason about the impact of gradient norms on the disparities in the group excessive losses.

**The role of gradient norms in pruning.** Having highlighted the connection between gradients norms of a group with the accuracy of the pruned model on such a group, this section provides theoretical intuitions on the role of gradient norms in the disparate group losses during pruning.

From Theorem 1, notice that the excessive loss is controlled by term $\|g_a^\ell\| \times \|\bar{\boldsymbol{\theta}} - \mathring{\boldsymbol{\theta}}\|$. As already noted in Corollary 1, the term $\|\bar{\boldsymbol{\theta}} - \mathring{\boldsymbol{\theta}}\|$ regulates the impact of pruning on the excessive loss, as the difference between the pruned and non-pruned parameters vectors directly depends on the pruning rate. For a fixed pruning rate, however, notice that groups with different gradient norms will have a disparate effect on the resulting term. In particular, groups with very small gradients norms (those generally associated with highly accurate predictions) will be less sensitive to the effects of the pruning rate. Conversely, groups with large gradient norms will be affected by the pruning rate to a greater extent, with larger pruning rates, *typically* reflecting in larger excessive losses.

These observations of the factors of disparity, accuracy, and group size, can also be appreciated empirically in Figures 4a and 4b. The plots report accuracy (a) and gradient norms (b) on the UTKFace datasets for a variety of pruning rates. Consider group *White* (containing 42% of the total samples) and *Others* (containing 7% of the total samples). The unpruned model has high accuracy on the former group and small gradient norms. The accuracy of this group is insensitive to various pruning rates and even increases at large pruning regimes. In contrast, group *Others* has much lower accuracy and larger gradient norms in the unpruned model. As the pruning rate increase, their accuracies drastically drop. As a result, in high pruning regimes, this minority group exhibits poor accuracy and very high gradient norms.

Notice that the empirical results apply to much more complex settings than those which can be analyzed formally, thus they complement the theoretical observations.

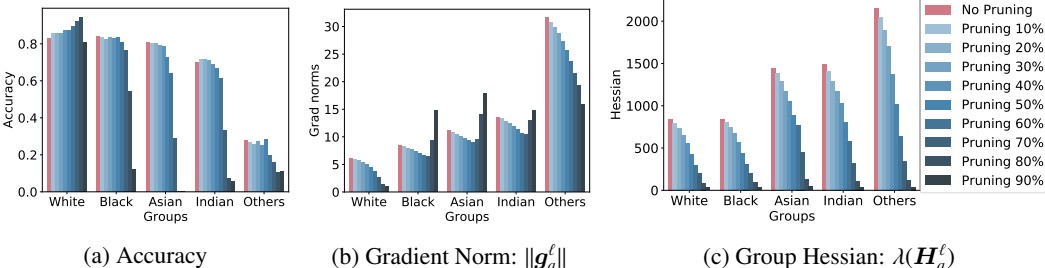

(a) Accuracy  (b) Gradient Norm: $\|g_a^\ell\|$  (c) Group Hessian: $\lambda(H_a^\ell)$

Figure 4: Accuracy, gradient norm, and group Hessian max eigenvalues of each ethnicity group, before and after increasing pruning ratios for UTK-Face dataset. The percentage of data samples across groups *White, Black, Asian, Indian, and Others* is $\sim 0.42, 0.19, 0.15, 0.15, 0.07$, respectively.

## 5 Why disparity in groups' Hessians causes unfairness?

Having examined the properties of the groups gradients and their relation to unfairness in pruning, this section turns on analyzing how the Hessian associated with the loss function for a group is linked to the unfairness observed during pruning. In more detail, it connects the groups' Hessian to the distance to the decision boundary for the samples in that group and their resulting model errors (Theorem 2), it illustrates a strong positive correlation between groups' Hessian and gradient norms, and links these concepts with the excessive loss (Theorem 1) to show that unfairness in model pruning is controlled by the difference in maximum eigenvalues of the Hessians among groups.

**Group Hessians and accuracy.** The section first shows that groups presenting large Hessian values may suffer larger disparate impacts due to pruning, when compared with groups that have smaller Hessians. It does so by connecting the maximum eigenvalues of the groups Hessians with their distance to decision boundary and the group accuracy. The following result sheds light on these observations. It restricts its attention to models trained under binary cross entropy losses, for clarity of explanation, although an extension to a multi-class case is directly attainable.

**Theorem 2.** *Let $f_\theta$ be a binary classifier trained using a binary cross entropy loss. For any group $a \in \mathcal{A}$, the maximum eigenvalue of the group Hessian $\lambda(H_a^\ell)$ can be upper bounded by:*

$$\lambda(H_a^\ell) \leq \frac{1}{|D_a|} \sum_{(x,y) \in D_a} \underbrace{\left(f_{\hat{\theta}}(x)\right)\left(1 - f_{\hat{\theta}}(x)\right)}_{\text{Closeness to decision boundary}} \times \left\|\nabla_\theta f_{\hat{\theta}}(x)\right\|^2 + \underbrace{\left|f_{\hat{\theta}}(x) - y\right|}_{\text{Error}} \times \lambda\left(\nabla_\theta^2 f_{\hat{\theta}}(x)\right). \quad (6)$$

The proof relies on derivations of the Hessian associated with model loss function and Weyl inequality. In other words, Theorem 2 highlights a direct connection between the maximum eigenvalue of the group Hessian and (**1**) the closeness to the decision boundary of the group samples, and (**2**) the accuracy of the group. The distance to the decision boundary is derived from [9]. Intuitively this term is maximized when the classifier is highly uncertain about the prediction: $f_{\hat{\theta}}(x) \to 0.5$, and minimized when it is highly certain $f_{\hat{\theta}}(x) \to 0$ or 1, as showed in the following proposition.

**Proposition 3.** *Consider a binary classifier $f_\theta(x)$. For a given sample $x \in D$, the term $f_{\hat{\theta}}(x)(1 - f_{\hat{\theta}}(x))$ is maximized when $f_{\hat{\theta}}(x) = 0.5$ and minimized when $f_{\hat{\theta}}(x) \in \{0, 1\}$.*

Observe that a group consisting of samples that are far from the decision boundary will have smaller Hessians and, thus, be less subject to a drop in accuracy due to model pruning. These results can be appreciated in Figure 5. Notice the inverse relationship between maximum eigenvalues of the groups' Hessians and their average distance to the decision boundary. The same relation also holds for accuracy: the higher the Hessians maximum eigenvalues, the smaller the accuracy. This is intuitive as samples which are close to the decision boundary will be more prone to errors due to small changes in the model due to pruning, when compared with samples lying far from the decision boundary.

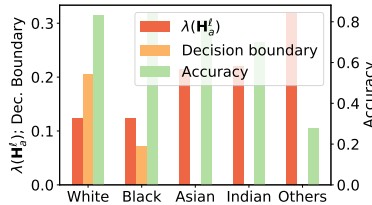

Figure 5: Group Hessians, distance to decision boundary, and accuracy.

**Correlation between group Hessians and gradient norms.** This section observes a positive correlation between maximum eigenvalues of the Hessian of a group and their gradient norms. This relation can be appreciated in Figure 6. While mainly empirical, this observation is important as it illustrates that both the Hessian $\lambda(H_a^\ell)$ and the gradient $\|g_a^\ell\|$ terms appearing in the upper bound of the excessive loss $R(a)$ reported in Theorem 1 are in agreement. This relation was observed in all our experiments and settings. Such observation allows us to infer that it is the combined effect of gradient norms and group Hessians that is responsible for the excessive loss of a group and, in turn, for the exacerbation of unfairness in the pruned models.

**The role of the group Hessian in pruning.** Having highlighted the connection between Hessian for a group with the resulting accuracy of the model on such a group, this section provides theoretical intuitions on the role of the Hessians in the disparate group losses during pruning.

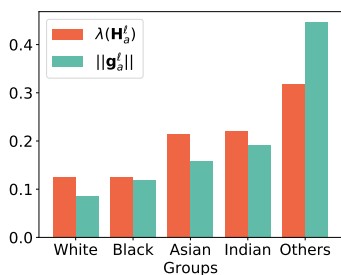

Figure 6: Group Hessians and gradient norms.

In Theorem 1, notice that the excessive loss is controlled by term $\|H_a^\ell\| \times \|\bar{\theta} - \overset{\star}{\theta}\|^2$. As also noted in the previous section, the term $\|\bar{\theta} - \overset{\star}{\theta}\|$ regulates the impact of pruning on the excessive loss as the difference between the pruned and non-pruned parameter vectors directly depends on the pruning rate. Similar to the observation for gradient norms, with a fixed pruning rate, groups with different Hessians will have a disparate effect on the resulting term. In particular, groups with small Hessians eigenvalues (those generally distant from the decision boundary and highly accurate) will be less sensitive to the effects of the pruning rate. Conversely, groups with large Hessians eigenvalues will be affected by the pruning rate to a greater extent, *typically* resulting in larger excessive losses. These observations can further be appreciated empirically in Figures 4a (for accuracy) and 4c (for maximum group Hessian eigenvalues) on the UTKFace datasets for a variety of pruning rates.

## 6 Mitigation solution and evaluation

The previous sections highlighted the presence of two key factors playing a role in the observed model accuracy disparities due to pruning: the difference in gradient norms, and the difference in Hessians losses across groups. This section first shows how to leverage these findings to provide a simple, yet effective solution to reduce the disparate impacts of pruning. Then, the section illustrates the benefits of this mitigating solution on a variety of tasks, datasets, and network architectures.

### 6.1 Mitigation solution

To achieve fairness, the aforementioned findings suggest to equalize the disparity associated with gradient norms $\|g_a^\ell\|$ and Hessians $\lambda(H_a^\ell)$ across different groups $a \in \mathcal{A}$. For this goal, we adopt a constrained empirical risk minimization approach:

$$\underset{\theta}{\text{minimize}} \quad J(\theta; D) \quad \text{such that: } \|g_a^\ell\| = \|g^\ell\|, \quad \lambda(H_a^\ell) = \lambda(H^\ell) \quad \forall a \in \mathcal{A}, \tag{7}$$

where $g^\ell = \nabla_\theta J(\theta; D)$ and $H^\ell = \nabla_\theta^2 J(\theta; D)$ refer to the gradients and Hessian associated with loss function $\ell$, respectively, and are computed using the whole dataset $D$. The approach (7) is a common strategy adopted in fair learning tasks, and the paper uses the Lagrangian Dual method of Fioretto et al. [12] which exploits Lagrangian duality to extend the loss function with trainable and weighted regularization terms that encapsulate constraints violations (see Appendix C for additional details).

A shortcoming of this approach is, however, that requires computing the gradient norms and Hessian matrices of the group losses in each and every training iteration, rendering the process computationally unviable, especially for deep, overparametrized networks.

To overcome this computational burden, we will use two observations made earlier in the paper. First, recall the strong relation between gradient norms for a group and their associated losses. This aspect was noted in Proposition 2. That is, when the losses across the groups are similar, the gradient norms across such groups will also tend to be similar. Next, Theorem 2 noted a positive correlation between model errors (and thus loss values) for a group and its associated Hessian eigenvalues. Thus, when the losses across the groups are similar, the group Hessians will also tend to be similar. This intuition

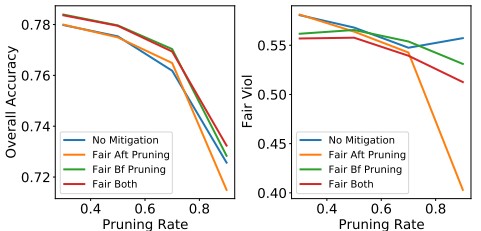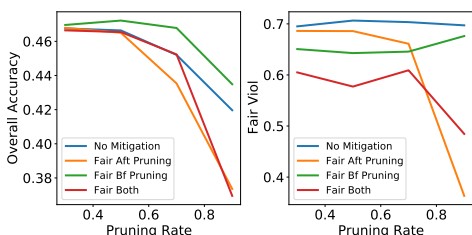

Figure 8: Accuracy and Fairness violations attained by all models on ResNet50, UTK-Face dataset with *ethnicity* (5 classes) as group attribute (and labels) [left] and *age* (9 classes) [right].

is also complemented by the strong correlation between group Hessians and gradient norms reported in Section 5. Based on the above observations, we propose a simpler version of the constrained minimizer (7) defined as

$$\underset{\boldsymbol{\theta}}{\text{minimize}} \quad J(\boldsymbol{\theta}; D) \quad \text{such that:} \quad J(\boldsymbol{\theta}; D_a) = J(\boldsymbol{\theta}; D) \quad \forall a \in \mathcal{A}, \tag{8}$$

that substitutes the gradient norms and max eigenvalues of group Hessians equality constraints with proxy terms capturing the group $J(\boldsymbol{\theta}; D_a)$ and population $J(\boldsymbol{\theta}; D)$ losses.

The impact of such proxy terms in the fairness-constrained problem above can be appreciated, empirically, in Figure 7. The plots, that use the UTK-Face dataset, with Ethnicity as protected group, show an original unfair model (top) and a fair counterpart obtained through problem (8) (bottom). Both top and bottom sub-figures use an unpruned model. The top sub-figure shows the performance of an original unpruned model trained by minimizing the empirical risk function while the bottom one shows the effect of solving Problem (8), i.e., it constrains the empirical risk function with the various group loss terms. Notice how enforcing balance in the group losses also helps reducing and balancing the gradient norms and group's average distance to the decision boundary. As a consequence, the resulting model fairness is dramatically enhanced (bottom-left subplot).

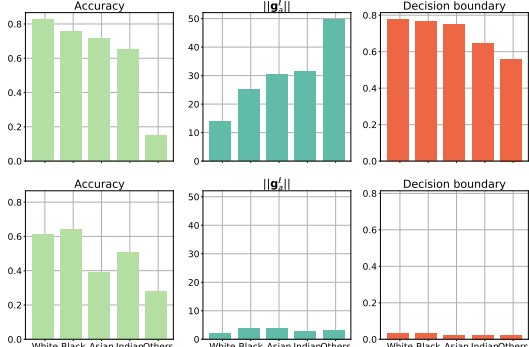

Figure 7: Effects of fairness constraints in balancing not only group accuracy (left) but also gradient norms (middle) and group average distance to the decision boundary (right).

## 6.2 Assessment of the mitigation solution

**Datasets, models, and settings.** This section analyzes the results obtained using the proposed mitigation solution with ResNet50 and VGG19 on the UTKFace dataset [41], CIFAR-10 [23], and SVHN [26] for various protected attributes. The experiments compare the following four models:

- *No Mitigation*: it refers to the standard pruning approach which uses no fairness mitigation strategy.
- *Fair Bf Pruning*: it applies the fairness mitigation process (Problem (8)) exclusively to the original large network, thus *before* pruning.
- *Fair Aft Pruning*: it applies the mitigation exclusively to the pruned network, thus *after* pruning.
- *Fair Both*: it applies the mitigation both to the original large network and to the pruned network.

The experiments report the overall accuracy of resulting models as well as their fairness violations, defined here as the difference between the maximal and minimal group accuracy. The reported metrics are the average of 10 repetitions. Additional details on datasets, architectures, and hyper-parameters adopted, as well as additional and extended results are reported in Appendix C.

**Effects on accuracy.** The section first focuses on analyzing the effects of accuracy drop due to applying the proposed mitigation solution for fair pruning. Figure 8 compares the four models on the UTK-Face dataset using a ResNet50 architecture. The left subplots use *ethnicity* as protected group and class label, with $|\mathcal{Y}| = 5$, while the right subplots use *age* as protected group and class label,

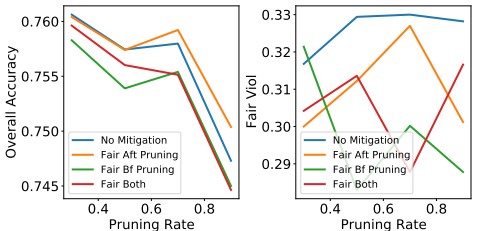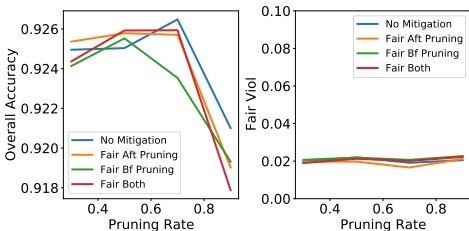

Figure 9: Accuracy and Fairness violations attained by all models on VGG-19, CIFAR-10 dataset (left) and SVHN (right) with 10 class labels also used as group attribute.

| Dataset | version | Class-wise accuracy | Overall accuracy |
|---|---|---|---|
| **UTK-age bins** | **full** | 0.856, 0.128, 0.145, 0.319, 0.331, 0.342, 0.181, 0.334, 0.512 | 0.395 |
| | **relaxed** | 0.810, 0.096, 0.141, 0.284, 0.385, 0.324, 0.227, 0.257, 0.533 | 0.390 |
| **UTK-gender** | **full** | 0.830, 0.876 | 0.857 |
| | **relaxed** | 0.868, 0.845 | 0.852 |
| **SVHN** | **full** | 0.864, 0.911, 0.869, 0.819, 0.887, 0.784, 0.840, 0.877, 0.805, 0.856 | 0.857 |
| | **relaxed** | 0.824, 0.910, 0.775, 0.726, 0.827, 0.752, 0.747, 0.789, 0.713, 0.755 | 0.795 |
| **MNIST** | **full** | 0.998, 0.996, 0.993, 0.998, 0.994, 0.991, 0.991, 0.993, 0.992, 0.985 | 0.993 |
| | **relaxed** | 0.994, 0.988, 0.989, 0.986, 0.987, 0.979, 0.981, 0.988, 0.969, 0.994 | 0.986 |

Table 1: Full (Equation 7) vs relaxed (Equation 8) versions of the proposed mitigation solutions.

with $|\mathcal{Y}| = 9$. Notice that, as expected, all compared models present some accuracy deterioration as the pruning rate increases. However, notably, the deterioration of the models that apply the fair mitigation steps are comparable to (or even improved) those of the "*No mitigation*" model, which applies standard pruning.

A similar trend can be seen in Figure 9 that reports results on CIFAR (left) and SVHN (right). Both use the ten class labels as protected attributes. These results clearly illustrate the ability of the mitigating solution to preserve highly accurate models.

A comparison of the "full" (Equation 7) and "relaxed" (Equation 8) versions of the proposed mitigation solutions is provided in Table 1. We note that while the "full" version leads to fairer results, the reduction in the various groups accuracy is often insubstantial. We also note that the running time of the "full" version is largely (over an order magnitude) longer than the relaxed counterpart. This is due to the calculation of gradient norms and the Hessian terms associated with each group.

**Effects on fairness.** The section next illustrates the ability of the proposed solution to achieve fair pruned models. Table 2 illustrates the results for the UTKFaces dataset with ethnicity as class labels and age as protected attributes for a CNN with two convolutional layers and three linear layers and prune amounts: 30%, 50%, 70%, and 90%. Notice how Fair Aft pruning and Fair both achieve relatively lesser fairness violations compared to the No mitigation and the Fair bf Pruning methods in most cases.

Next, the second and fourth subplots presented in Figures 8 and 9 illustrate the fairness violations obtained by the four models analyzed on different datasets and settings. We make the following observations: First, all the plots exhibit a consistent trend in that the mitigation solution produces models which improve the fairness of the baseline, "*No mitigation*" model. Observe that, as already illustrated in Figure 7, the fair models tend to equalize the gradient norms and group Hessians components (and thus the distance to the decision boundary across groups). Thus, the resulting pruned models also attain better fairness, when compared to their standard counterparts.

Next, notice that "*Fair Aft Pruning*" often achieves better fairness violations than "*Fair Bf Pruning*", especially at high pruning regimes. This is because the former has the advantage to apply the mitigation solution directly to the pruned model to ensure that the resulting model has low differences in gradient norms and group Hessians. The presentation also illustrates the application of the mitigation strategies both before and after pruning (*Fair Both*) which shows once again the significance of applying the mitigation solution over the pruned network.

| Methods | Overall accuracy | | | | Fairness violations | | | |
|---|---|---|---|---|---|---|---|---|
| | 30% | 50% | 70% | 90 % | 30% | 50% | 70% | 90% |
| **No mitigation** | 0.546 | 0.545 | 0.529 | 0.559 | 0.179 | 0.186 | 0.152 | 0.134 |
| **Fair bf Pruning** | 0.539 | 0.557 | 0.529 | 0.540 | 0.189 | 0.190 | 0.174 | 0.238 |
| **Fair Aft Pruning** | 0.538 | 0.532 | 0.497 | 0.472 | 0.172 | 0.161 | 0.163 | 0.05 |
| **Fair both** | 0.525 | 0.541 | 0.508 | 0.484 | 0.170 | 0.144 | 0.156 | 0.073 |

Table 2: Accuracy and fairness violations for the UTKFaces dataset with *ethnicity* as class labels and *age* as protected attributes and prune amounts of 30%, 50%, 70%, and 90%.

Finally, it is notable that "*Fair Aft Pruning*" achieves good reductions in fairness violation. Indeed, pre-trained large (non-pruned) fair models may not be available and the ability to retrain these large models prior to pruning may be hindered by their size and complexity.

## 7   Discussion and limitations

This section discusses three key messages found in this study. First, we notice that pruning affecting model separability and distance to the decision boundary is related to concepts also explored in robust machine learning [15, 28]. Not surprisingly, some recent literature in network pruning has empirically observed that pruning may have a negative impact on adversarial robustness [16]. These observations raise questions about the connection between pruning, robustness, and fairness, which we believe is an important direction to further investigate.

Next, although the solution proposed in Problem (8) allows it to be adopted in large models, the size of modern ML models (together with the amount of hyperparameters searches) may hinder retraining such original massive models from incorporating fairness constraints. Notably, however, the proposed mitigation solution can be used as a post-processing step to be applied during the pruning operation directly. The previous section shows that the proposed method delivers desirable performance in terms of both accuracy and fairness.

Finally, we notice that the results analyzed in this paper pertain to losses that are twice differentiable. Lifting such an assumption will be an interesting and challenging future research avenue.

**Ethical considerations.**   The analyses and solutions reported in this paper should not be intended as an endorsement for using the developed techniques to aid facial recognition systems. We hope this work creates further awareness of the unfairness caused by pruning mechanisms in ML systems in the context of models that could be deployed in energy-efficient devices, such as smart cameras or access control systems, etc.

## 8   Conclusion

This work observed that pruning, while effective in compressing large models with minimal loss of accuracy, can result in substantial disparate accuracy impacts. The paper examined the factors causing such disparities both theoretically and empirically showing that: **(1)** disparity in gradient norms across groups and **(2)** disparity in Hessian matrices associated with the loss functions computed using a groups' data are two key factors responsible for such disparities to arise. By recognizing these factors, the paper also developed a simple yet effective retraining technique that largely mitigates the disparate impacts caused by pruning.

As reduced versions of large, overparameterized models become increasingly adopted in embedded systems to facilitate autonomous decisions, we believe that this work makes an important step toward *understanding* and *mitigating* the sources of disparate impacts observed in compressed learning models and hope it will spark awareness in this important area.

## Acknowledgments

This research is partly funded by NSF grants SaTC-1945541, SaTC-2133169, and CAREER-2143706. F. Fioretto is also supported by a Google Research Scholar Award and an Amazon Research Award. The views and conclusions of this work are those of the authors only.

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
