# A Missing Proofs

**Theorem 1.** *The excessive loss of a group $a \in \mathcal{A}$ is upper bounded by[3]:*

$$R(a) \leq \|g_a^\ell\| \times \|\bar{\theta} - \mathring{\theta}\| + \frac{1}{2}\lambda\left(H_a^\ell\right) \times \|\bar{\theta} - \mathring{\theta}\|^2 + O\left(\|\bar{\theta} - \mathring{\theta}\|^3\right), \tag{9}$$

*where $g_a^\ell = \nabla_{\mathring{\theta}} J(\mathring{\theta}; D_a)$ is the vector of gradients associated with the loss function $\ell$ evaluated at $\mathring{\theta}$ and computed using group data $D_a$, $H_a^\ell = \nabla_{\mathring{\theta}}^2 J(\mathring{\theta}; D_a)$ is the Hessian matrix of the loss function $\ell$, at the optimal parameters vector $\mathring{\theta}$, computed using the group data $D_a$ (henceforth simply referred to as group hessian), and $\lambda(\Sigma)$ is the maximum eigenvalue of a matrix $\Sigma$.*

*Proof.* Using a second order Taylor expansion around $\mathring{\theta}$, the excessive loss $R(a)$ for a group $a \in \mathcal{A}$ can be stated as:

$$R(a) = J(\bar{\theta}; D_a) - J(\mathring{\theta}; D_a)$$

$$= \left[J\left(\mathring{\theta}; D_a\right) + \left(\bar{\theta} - \mathring{\theta}\right)^\top \nabla_\theta J\left(\mathring{\theta}; D_a\right) + \frac{1}{2}\left(\bar{\theta} - \mathring{\theta}\right)^\top H_a^\ell\left(\bar{\theta} - \mathring{\theta}\right) + O\left(\|\mathring{\theta} - \bar{\theta}\|^3\right)\right] - J\left(\mathring{\theta}; D_a\right)$$

$$= \left(\bar{\theta} - \mathring{\theta}\right)^\top g_a^\ell + \frac{1}{2}\left(\bar{\theta} - \mathring{\theta}\right)^\top H_a^\ell\left(\bar{\theta} - \mathring{\theta}\right) + O\left(\|\mathring{\theta} - \bar{\theta}\|^3\right)$$

The above, follows from the loss $\ell(\cdot)$ being at least twice differentiable, by assumption.

By Cauchy-Schwarz inequality, it follows that

$$\left(\bar{\theta} - \mathring{\theta}\right)^\top g_a^\ell \leq \|\bar{\theta} - \mathring{\theta}\| \times \|g_a^\ell\|.$$

In addition, due to the property of Rayleigh quotient we have:

$$\frac{1}{2}\left(\bar{\theta} - \mathring{\theta}\right)^\top H_a^\ell\left(\bar{\theta} - \mathring{\theta}\right) \leq \frac{1}{2}\lambda\left(H_a^\ell\right) \times \|\bar{\theta} - \mathring{\theta}\|^2.$$

The upper bound for the excessive loss $R(a)$ is thus obtained by combining these two inequalities. $\square$

**Proposition 1.** *Consider two groups $a$ and $b$ in $\mathcal{A}$ with $|D_a| \geq |D_b|$. Then $\|g_a^\ell\| \leq \|g_b^\ell\|$.*

*Proof.* By the assumption that the model converges to a local minima, it follows that:

$$\nabla_\theta \mathcal{L}(\mathring{\theta}; D) = \sum_{a \in \mathcal{A}} \frac{|D_a|}{|D|} \nabla_\theta J(\mathring{\theta}; D_a)$$

$$= \frac{|D_a|}{|D|} g_a^\ell + \frac{|D_b|}{|D|} g_b^\ell = 0$$

Thus, $g_a^\ell = -\frac{|D_b|}{|D_a|} g_b$. Hence $\|g_a^\ell\| = \frac{|D_b|}{|D_a|}\|g_b^\ell\| \leq \|g_b^\ell\|$, because $|D_a| \geq |D_b|$. $\square$

**Proposition 2.** *For a given group $a \in \mathcal{A}$, gradient norms can be upper bounded as:*

$$\|g_a^\ell\| \in O\left(\sum_{(\boldsymbol{x}, y) \in D_a} \underbrace{\|f_{\mathring{\theta}}(\boldsymbol{x}) - y\|}_{Accuracy} \times \left\|\nabla_{\mathring{\theta}} f_{\mathring{\theta}}(\boldsymbol{x})\right\|\right).$$

The above proposition is presented in the context of cross entropy loss or mean squared error loss functions. These two cases are reviewed as follows

---

[3]With a slight abuse of notation, the results refer to $\bar{\theta}$ as the homonymous vector which is extended with $k - \bar{k}$ zeros.

**Cross Entropy Loss.** Consider a classification task with cross entropy loss: $\ell(f_{\hat{\theta}}(\boldsymbol{x}), y) = -\sum_{z \in \mathcal{Y}} f_{\hat{\theta}}^{z}(\boldsymbol{x}) \boldsymbol{y}^{z}$, where $f_{\hat{\theta}}^{z}(\boldsymbol{x})$ represents the $z$-th element of the output associated with the softmax layer of model $f_{\hat{\theta}}$, and $\boldsymbol{y}$ is a one-hot encoding of the true label $y$, with $\boldsymbol{y}^z$ representing its $z$-th element, then,

$$\|g_a\| = \left\|\nabla_{\boldsymbol{\theta}} J(\dot{\boldsymbol{\theta}}; D_a,)\right\| = \left\|\frac{1}{|D_a|} \sum_{(\boldsymbol{x},y) \in D_a} \nabla_f \ell(f_{\hat{\theta}}(\boldsymbol{x}), y) \times \nabla_{\boldsymbol{\theta}} f_{\hat{\theta}}(\boldsymbol{x})\right\|$$

$$= \left\|\frac{1}{|D_a|} \sum_{(\boldsymbol{x},y) \in D_a} (f_{\hat{\theta}}(\boldsymbol{x}) - \boldsymbol{y}) \times \nabla_{\boldsymbol{\theta}} f_{\hat{\theta}}(\boldsymbol{x})\right\|$$

$$\leq \frac{1}{|D_a|} \sum_{(\boldsymbol{x},y) \in D_a} \left\|f_{\hat{\theta}}(\boldsymbol{x}) - \boldsymbol{y}\right\| \times \left\|\nabla_{\boldsymbol{\theta}} f_{\hat{\theta}}(\boldsymbol{x})\right\|,$$

where the third equality is due to that the gradient of the cross entropy loss reduces to $f_{\hat{\theta}}(\boldsymbol{x}) - \boldsymbol{y}$.

**Mean Squared Error.** Next, consider a regression task with mean squared error loss $\ell(f_{\hat{\theta}}(\boldsymbol{x}), y) = (f_{\hat{\theta}}(\boldsymbol{x}) - y)^2$. Using the same notation as that made above, if follows:

$$\|g_a\| = \left\|\nabla_{\boldsymbol{\theta}} J(\dot{\boldsymbol{\theta}}; D_a,)\right\| = \left\|\frac{1}{|D_a|} \sum_{(\boldsymbol{x},y) \in D_a} \nabla_f \ell(f_{\hat{\theta}}(\boldsymbol{x}), y) \times \nabla_{\boldsymbol{\theta}} f_{\hat{\theta}}(\boldsymbol{x})\right\|$$

$$= \left\|\frac{2}{|D_a|} \sum_{(\boldsymbol{x},y) \in D_a} (f_{\hat{\theta}}(\boldsymbol{x}) - \boldsymbol{y}) \times \nabla_{\boldsymbol{\theta}} f_{\hat{\theta}}(\boldsymbol{x})\right\|$$

$$\leq \frac{2}{|D_a|} \sum_{(\boldsymbol{x},y) \in D_a} \left\|f_{\hat{\theta}}(\boldsymbol{x}) - \boldsymbol{y}\right\| \times \left\|\nabla_{\boldsymbol{\theta}} f_{\hat{\theta}}(\boldsymbol{x})\right\|,$$

where the third equality is due to that the gradient of the mean squared error loss w.r.t. $f_{\hat{\theta}}(\cdot)$ reduces to $2(f_{\hat{\theta}}(\boldsymbol{x}) - \boldsymbol{y})$.

**Theorem 2.** *Let $f_\theta$ be a binary classifier trained using a binary cross entropy loss. For any group $a \in \mathcal{A}$, the maximum eigenvalue of the group Hessian $\lambda(\boldsymbol{H}_a^\ell)$ can be upper bounded by:*

$$\lambda(\boldsymbol{H}_a^\ell) \leq \frac{1}{|D_a|} \sum_{(\boldsymbol{x},y) \in D_a} \underbrace{\left(f_{\hat{\theta}}(\boldsymbol{x})\right)\left(1 - f_{\hat{\theta}}(\boldsymbol{x})\right)}_{\text{Distance to decision boundary}} \times \left\|\nabla_{\boldsymbol{\theta}} f_{\hat{\theta}}(\boldsymbol{x})\right\|^2 + \underbrace{\left|f_{\hat{\theta}}(\boldsymbol{x}) - y\right|}_{\text{Accuracy}} \times \lambda\left(\nabla_{\boldsymbol{\theta}}^2 f_{\hat{\theta}}(\boldsymbol{x})\right). \quad (10)$$

*Proof.* First notice that an upper bound for the Hessian loss computed on a group $a \in \mathcal{A}$ can be derived as:

$$\lambda(\boldsymbol{H}_a^\ell) = \lambda\left(\frac{1}{|D_a|} \sum_{(\boldsymbol{x},y) \in D_a} \boldsymbol{H}_{\boldsymbol{x}}^\ell\right) \leq \frac{1}{|D_a|} \sum_{(\boldsymbol{x},y) \in D_a} \lambda\left(\boldsymbol{H}_{\boldsymbol{x}}^\ell\right) \quad (11)$$

where $\boldsymbol{H}_{\boldsymbol{x}}^\ell$ represents the Hessian loss associated with a sample $\boldsymbol{x} \in D_a$ from group $a$. The above follows Weily's inequality which states that for any two symmetric matrices $A$ and $B$, $\lambda(A + B) \leq \lambda(A) + \lambda(B)$.

Next, we will derive an upper bound on the Hessian loss associated to a sample $\boldsymbol{x}$. First, based on the chain rule a closed form expression for the Hessian loss associated to a sample $\boldsymbol{x}$ can be written as follows:

$$\boldsymbol{H}_{\boldsymbol{x}}^\ell = \nabla_f^2 \ell\left(f_{\hat{\theta}}(\boldsymbol{x}), y\right)\left[\nabla_{\boldsymbol{\theta}} f_{\hat{\theta}}(\boldsymbol{x})\left(\nabla_{\boldsymbol{\theta}} f_{\hat{\theta}}(\boldsymbol{x})\right)^\top\right] + \nabla_f \ell\left(f_{\hat{\theta}}(\boldsymbol{x}), y\right) \nabla_{\boldsymbol{\theta}}^2 f_{\hat{\theta}}(\boldsymbol{x}). \quad (12)$$

The next follows from that

$$\nabla_f \ell\left(f_{\hat{\theta}}(\boldsymbol{x}), y\right) = (f_{\hat{\theta}}(\boldsymbol{x}) - y),$$

$$\nabla_f^2 \ell\left(f_{\hat{\theta}}(\boldsymbol{x}), y\right) = f_{\hat{\theta}}(\boldsymbol{x})\left(1 - f_{\hat{\theta}}(\boldsymbol{x})\right).$$

Applying the Weily inequality again on the R.H.S. of Equation 12, we obtain:

$$\lambda(\boldsymbol{H}_{\boldsymbol{x}}^{\ell}) \leq f_{\hat{\boldsymbol{\theta}}}(\boldsymbol{x})\left(1 - f_{\hat{\boldsymbol{\theta}}}(\boldsymbol{x})\right) \times \left\|\nabla_{\boldsymbol{\theta}} f_{\hat{\boldsymbol{\theta}}}(\boldsymbol{x})\right\|^2 + \lambda\left(f_{\hat{\boldsymbol{\theta}}}(\boldsymbol{x}) - y\right) \times \nabla_{\boldsymbol{\theta}}^2 f_{\hat{\boldsymbol{\theta}}}(\boldsymbol{x})$$

$$\leq f_{\hat{\boldsymbol{\theta}}}(\boldsymbol{x})\left(1 - f_{\hat{\boldsymbol{\theta}}}(\boldsymbol{x})\right) \times \left\|\nabla_{\boldsymbol{\theta}} f_{\hat{\boldsymbol{\theta}}}(\boldsymbol{x})\right\|^2 + \left|f_{\hat{\boldsymbol{\theta}}}(\boldsymbol{x}) - y\right| \lambda\left(\nabla_{\boldsymbol{\theta}}^2 f_{\hat{\boldsymbol{\theta}}}(\boldsymbol{x})\right) \tag{13}$$

The statement of Theorem 2 is obtained combining Equations 13 with 11. $\qquad\square$

**Proposition 3.** *Consider a binary classifier $f_{\boldsymbol{\theta}}(\boldsymbol{x})$. For a given sample $\boldsymbol{x} \in D$, the term $f_{\hat{\boldsymbol{\theta}}}(\boldsymbol{x})(1 - f_{\hat{\boldsymbol{\theta}}}(\boldsymbol{x}))$ is maximized when $f_{\hat{\boldsymbol{\theta}}}(\boldsymbol{x}) = 0.5$ and minimized when $f_{\hat{\boldsymbol{\theta}}}(\boldsymbol{x}) \in \{0, 1\}$.*

*Proof.* First, notice that $f_{\hat{\boldsymbol{\theta}}}(\boldsymbol{x}) \in [0, 1]$, as it represents the soft prediction (that returned by the last layer of the network), thus $f_{\hat{\boldsymbol{\theta}}}(\boldsymbol{x}) \geq f_{\hat{\boldsymbol{\theta}}}^2(\boldsymbol{x})$. It follows that:

$$f_{\hat{\boldsymbol{\theta}}}(\boldsymbol{x})\left(1 - f_{\hat{\boldsymbol{\theta}}}(\boldsymbol{x})\right) = f_{\hat{\boldsymbol{\theta}}}(\boldsymbol{x}) - f_{\hat{\boldsymbol{\theta}}}^2(\boldsymbol{x}) \geq 0. \tag{14}$$

In the above, it is easy to observe that the equality holds when either $f_{\hat{\boldsymbol{\theta}}}(\boldsymbol{x}) = 0$ or $f_{\hat{\boldsymbol{\theta}}}(\boldsymbol{x}) = 1$.

Next, by the Jensen inequality, it follows that:

$$f_{\hat{\boldsymbol{\theta}}}(\boldsymbol{x})\left(1 - f_{\hat{\boldsymbol{\theta}}}(\boldsymbol{x})\right) \leq \frac{\left(f_{\hat{\boldsymbol{\theta}}}(\boldsymbol{x}) + 1 - f_{\hat{\boldsymbol{\theta}}}(\boldsymbol{x})\right)^2}{4} = \frac{1}{4}. \tag{15}$$

The above holds when $f_{\hat{\boldsymbol{\theta}}}(\boldsymbol{x}) = 1 - f_{\hat{\boldsymbol{\theta}}}(\boldsymbol{x})$, in other words, when $f_{\hat{\boldsymbol{\theta}}}(\boldsymbol{x}) = 0.5$. Notice that, in the case of binary classifier, this refers to the case when the sample $\boldsymbol{x}$ lies on the decision boundary. $\qquad\square$

## B  Dataset and Experimental Settings

### B.1  Datasets

The paper uses the following datasets to validate the findings discussed in the main paper:

- **UTK-Face** [41]. A large-scale face dataset with a long age span (range from 0 to 116 years old). The dataset consists of over 20,000 face images with annotations of age, gender, and ethnicity. The images cover large variations in pose, facial expression, illumination, occlusion, resolution, etc. The experiments adopt the following attributes for classification (e.g., $\mathcal{Y}$) and as protected group ($\mathcal{A}$): *ethnicity, age bins, gender*.

- **CIFAR-10** [23]. This dataset consists of 60,000 32×32 RGB images in 10 classes, with 6,000 images per class. The 10 different classes represent airplanes, cars, birds, cats, deer, dogs, frogs, horses, ships, and trucks.

- **SVHN** [26] Street View House Numbers (SVHN) is a digit classification dataset that contains 600,000 32×32 RGB images of printed digits (from 0 to 9) cropped from pictures of house number plates.

### B.2  Architectures, Hyper-parameters, and Settings

The study adopts the following architectures to validate the results of the main paper:

- **ResNet18**: An 18-layer architecture, with 8 residual blocks. Each residual block consists of two convolution layers. The model has $\sim$ 11 million trainable parameters.

- **ResNet50** This model contains 48 convolution layers, 1 MaxPool layer and a AvgPool layer. ResNet50 has $\sim$ 25 million trainable parameters.

- **VGG-19** This model consists of 19 layers (16 convolution layers, 3 fully connected layers, 5 MaxPool layers and 1 SoftMax layer). The model has $\sim$ 143 million parameters.

Hyperparameters for each of the above models was performed over a grid search (for different learning rates $= [0.0001, 0.001, 0.01, 0.1, 0.5, 0.05, 0.005, 0.0005]$) over a cluster of NVIDIA RTX A6000 with the above networks using the UTKFace dataset. The models with the highest accuracy

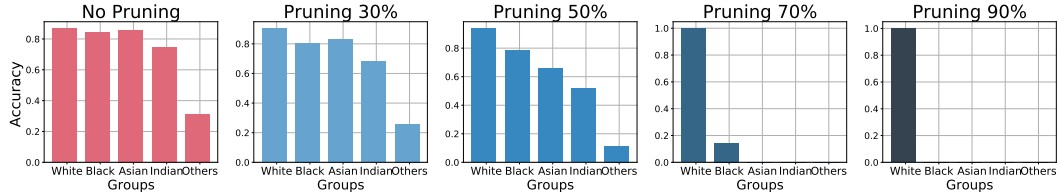

Figure 10: Accuracy of each demographic group in the UTK-Face dataset with ethnicity (5 classes) as group attribute using VGG19 over increasing pruning rates.

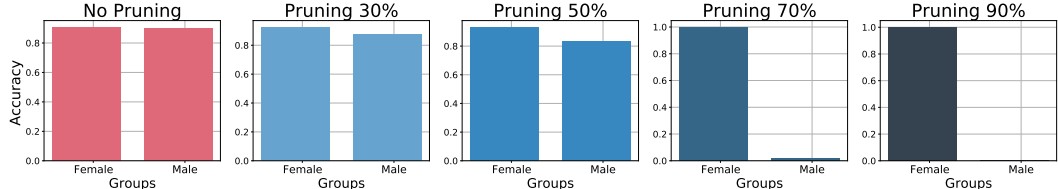

Figure 11: Accuracy of each demographic group in the UTK-Face dataset with gender (2 classes) as group attribute using VGG19 over increasing pruning rates.

were chosen and employed for the assessment of the mitigation solution in Sec. 6.2. The running time required for all sets of experiments which include mitigation solutions was about ~3 *days*.

The training uses the SGD optimizer with a momentum of 0.9 and weight_decay of $1e^{-4}$. Finally, the Lagrangian step size adopted in the Lagrangian dual learning framework [12] is set to 0.001.

All the models developed were implemented using Pytorch 3.0. The training was performed using NVidia Tesla P100-PCIE-16GB GPUs and 2GHz Intel Cores. The model was run for 100 epochs for the CIFAR-10 and SVHN and 40 epochs for UTK-Face dataset. Each reported experiment is an average of 10 repetitions. In all experiments, the protected group set coincides with the target label set: i.e., $\mathcal{A} = \mathcal{Y}$.

## C   Additional Experimental Results

### C.1   Impact of pruning on fairness

This section shows and affirms the impact of pruning towards accuracy disparity through VGG-19 network. The same training procedures as employed with ResNet18 in Fig 1 were followed. Each demographic group's accuracy is shown before and after pruning on the UTK-Face dataset in two cases: when *ethnicity* is a group attribute as in Figure 10, and when *gender* is a group attribute as in Figure 11. A consistent message is that under a higher pruning rate, the accuracies are more imbalanced across different groups, emphasizing the negative impact of pruning on fairness.

### C.2   Correlation of gradient/hessian norm and average distance to the decision boundary

This subsection elaborates the impact of gradient norms and group Hessians towards the fairness issues shown in Figures 10 and 11. In Section 4, it has been shown that the group with a larger gradient norm before pruning will be penalized more than the groups with a smaller gradient norm. Figures 13 and 12 show the gradient norm of each demographic group for UTK-Face dataset under two choices of protected attributes for VGG 19 networks. The results indicate that a group penalized less will have a smaller gradient norm with respect to those of the other groups.

In addition, Section 5 supports that Hessian norm is another factor. More precisely, the groups with a larger Hessian norm will be penalized more (drop much more in accuracy) than groups with a smaller Hessian norm. Evidence is provided for the claim on VGG19 in Figures 12 and 13. These results on VGG19 again confirm the theoretical findings.

Finally, in Section 5, a positive correlation between gradient norms and Hessian groups is shown in Theorem 2, and a negative correlation between Hessian groups and distance to the decision boundary

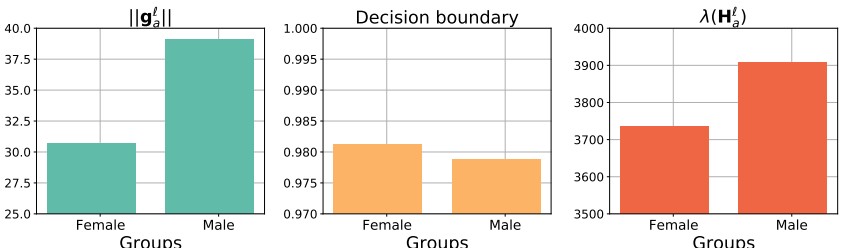

Figure 12: Gradient/Hessian norm and average distance to the decision boundary of each demographic group in the UTK-Face dataset with gender (2 classes) as group attribute using VGG19 with no pruning.

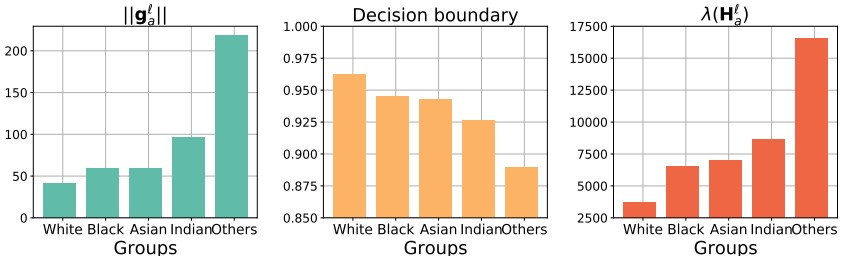

Figure 13: Gradient/Hessian norm and average distance to the decision boundary of each demographic group in the UTK-Face dataset with ethnicity (5 classes) as group attribute using VGG19 with no pruning.

is shown in Proposition 3. These important results again are supported by the results in Figures 12 and 13.

### C.3 Impact of group sizes to gradient norm

This section presents additional empirical results to support Theorem 1, stating that the group with more samples will tend to have a smaller gradient norm. In these experiments, run on a ResNet50 network, one group is chosen and upsampled $1\times$, $5\times$, $10\times$, and $20\times$ times. Note that by increasingly upsampling it, the group becomes the majority group in that dataset. A group with *more samples* is expected to end up with a *smaller gradient norm* when the training convergences.

**UTK-Face with gender** Since the UTK-Face is balanced with regard to gender (Female/Male), the number of samples in Female, and Male groups is upsampled in turn. Figure 14 reports the respective gradient norms at the last training iteration when upsampling Females (left) and Males (right.) Note how the Male group, initially with no upsampling, has a larger gradient norm than the Female group (right sub-plot). However, if the number of Male samples is increased enough, its gradient norm becomes smaller than that of the Female group.

**UTK-Face with age bins** Similar experiments are performed with UTK-Face on nine age bin groups. Three age bins are randomly chosen, 0, 2, 4, and the number of samples for each group is upsampled in turn. The gradient norms of nine age bin groups are shown in Figure 15, where the upsampled groups are highlighted with dotted thick lines. The results echo that if a group's number of samples is increased enough, its gradient norm at convergence will be smaller than the other 8 age bin groups.

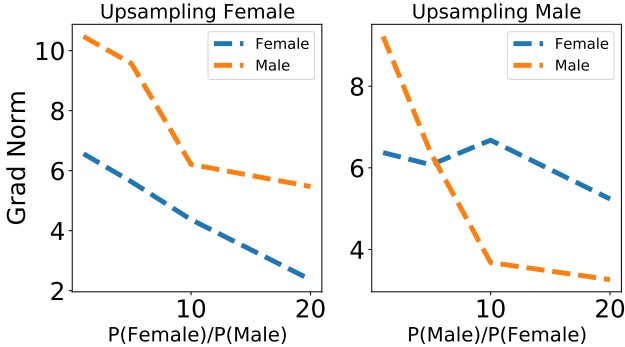

Figure 14: Impact of group sizes to the gradient norm per group in UTK-Face datase where groups are Male and Female.

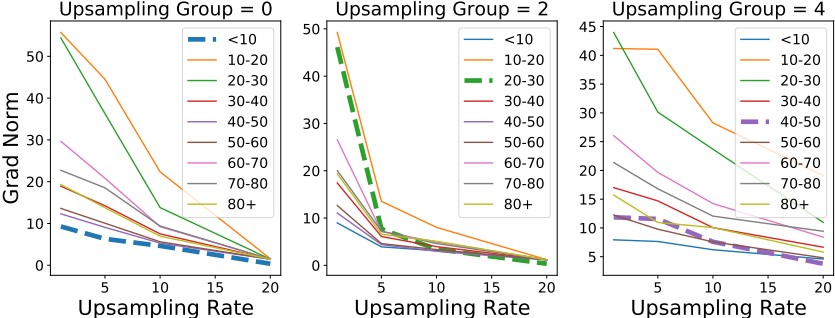

Figure 15: Impact of group sizes to the gradient norm per group in UTK-Face dataset where groups are nine age bins. The group with dotted thick line is a *majority group* in each chart.