# OpenReview forum: "Pruning has a disparate impact on model accuracy"
_NeurIPS.cc/2022/Conference — NeurIPS 2022 Accept_

### Official Review · Reviewer_B2Lo · 2022-07-11

**Rating:** 7
**Confidence:** 3
**Soundness:** 3 good
**Presentation:** 4 excellent
**Contribution:** 4 excellent

**Summary:**

This paper studies the relationship between pruning and group disparities in model accuracy. Using the UTKFace dataset and ResNet and VGG models, the authors show that increased levels of pruning show an increase in disparities in accuracy between groups. That is, groups that had lower accuracy than average had their accuracies get lower, while groups with higher than average accuracy have their accuracies improve. The authors relate these changes theoretically to the gradient norm and Hessian and propose mitigation techniques with these insights.

**Questions:**

In equation 3, why is the maximum chosen for fairness violation? Was the average, standard deviation, etc. also considered? Does max have desirable theoretical properties that the others don’t? Using the maximum implies a worst case fairness violation, but it could be interesting to understand the behavior of, say, the average fairness violation as well. This may be worth discussing in the text.

Rather than having the two bars per group in figure 2, would it be possible to made a 2D plot of group size vs gradient norm instead? For this figure, it’s not crucial to know which groups are which, but instead to see the correlation between group size and gradient norm more directly. It can be inferred from the bars, but a scatter plot might convey the results more clearly.

Given that a relationship between gradient norms and loss is shown, what is the relationship between fairness violation and accuracy? Is there an inherent tradeoff? In interpreting the results of figure 9, I found myself wondering whether having a lower accuracy automatically meant having a higher fairness violation. It was hard to tell from the plots themselves, but I think this is worth some discussion.


**Limitations:**

The primary limitation of the work is that it only considers pruning in the context of a single dataset and task (with multiple models). So, some of the empirical results may not generalize to, say, a BERT model performing sentiment classification. That being said, the theoretical results are not task dependent.

**Strengths And Weaknesses:**

The paper’s main strengths are its originality, clarity, and quality. The effect of pruning on group disparity metrics has not been studied in the past, and this paper does a very thorough theoretical and empirical treatment of the effect. In particular, the insights about the relationship of the gradient norm and Hessian eigenvalues to fairness violation were well done. The paper is systematic in its approach and exposition, with an easy-to-follow motivation.

The main weakness of the paper is that the mitigation results presented are difficult to interpret. While the results showing the relationship between pruning and fairness violation are quite clear, the results shown in figure 9 are difficult to read and understand. For example, consider the fairness violation plot in the bottom left of Figure 9. Here there is no smooth behavior relative to the pruning fraction, despite the fact that the theoretical results seem to indicate that there should be. It looks as though there may be some statistical uncertainty in these experiment results.

Even without these mitigation results, however, the theoretical treatment and empirical results on pruning are interesting and could be a paper on their own.

---

> ### Author Response · Authors · 2022-08-01
> **Authors' response**
>
> We thank the reviewer for providing valuable feedback. We are happy our work was well received; we also believe this is an important area that deserves attention. We answer your questions below, and we will be happy to respond to other questions/doubts you may have about this work.
>
> **Q1**
>
> Maximum violation provides a worst-case fairness violation, as you had suggested. And we adopt this definition of fairness throughout our paper as it quantifies the **worst** case of violating fairness in comparison to an average case. We agree that the average and standard deviation might also offer insights on fairness issues, and we will recommend it as an avenue for future work in our final version.
>
> **Q2**
>
> Thank you for the suggestion! We believe this suggestion can make a good replacement for the bar plot of Figure 2. We will update the Figure in our final version.
>
> **Q3**
>
> Thank you for this question! As you said, Figure 9 does imply lower accuracy leads to lower fairness violation. In general, there should NOT be any dependency between accuracy and fairness violation. However, there might be a correlation between them.
> When `overall accuracy` is lower,  the `maximum accuracy` is lower and when the `overall accuracy` approaches zero, then `max accuracy` and `min accuracy` of individual classes also approach zero.
> By the definition of fairness violation,
> `fairness violation = max acc - min acc` will approach zero as well.
> Hence, there might be some trends that denote low accuracy leading to lower fairness violations depending on the dataset, but these may not hold in general.
>
> We hope to have provided clarifications to your doubts and are happy to discuss any further questions

---

> > ### Comment · Reviewer_B2Lo · 2022-08-03
> > **Reviewer response**
> >
> > Thank you for your clarifications to the questions! Your answers make sense. I am glad to see the suggestion in Q2 was well-received, and I think adding some of the responses to the other questions to the paper text will improve it as well

---

> > > ### Author Response · Authors · 2022-08-04
> > > **Re: Reviewer response**
> > >
> > > Many thanks!  We agree. We will add a comment about `Q1` in our discussion section (Section 7) and a comment related to our answer to  `Q3` in Section 6.2, when discussing the assessment of the mitigating solution.
> > >
> > > Once again, we are glad our paper was well-received and are happy to discuss any further questions!

---

### Official Review · Reviewer_3Nrs · 2022-07-12

**Rating:** 6
**Confidence:** 4
**Soundness:** 3 good
**Presentation:** 3 good
**Contribution:** 3 good

**Summary:**

This paper has studied two factors causing disparity of network pruning, that is differences of gradient norm and Hessian matrix cross groups. It further introduced a simple training strategy to mitigate this disparity. With extensive experiments on UTK-Face, CIFAR-10, and SVH datasets,  this strategy has shown to mitigate the disparity well and keep comparable pruning performance to those without migration.

**Questions:**

1. It is unclear to me how to do the pruning with mitigation. Does the mitigation after pruning need retraining the model?
2. In Figure 7, although the fairness among groups is improved,  the pruning accuracy of all groups except “Others” equally decreases with that. The unfair model (top) seems to be an unpruned model. Is the fair model (bottom) the one before pruning or after pruning? If it is after pruning, please clarify the pruning rate.
3. If I understand correctly, all the experiments have the class labels as the protected groups. How does this mitigation strategy work when they are different? For example, the model on the UTK-Face dataset uses ethnicity as class labels and uses age as protected groups.
4. In Proposition 2 and Theorem 2, $||f_{\theta}(x) -y||$ is for the error instead of the accuracy, $(f_\theta(x)(1-f_\theta(x)))$ and the distance to the decision boundary seems negative proportional. The larger  $(f_\theta(x)(1-f_\theta(x)))$, the smaller the distance to the decision boundary is.
5. In line 94, what is the soft output? Is it the output after softmax or before it?

**Limitations:**

The proposed mitigation solution uses the loss disparity cross groups as the fairness proxy. Although much simpler, it is not so novel that it can be proposed without knowing the gradient norms and Hessian matrices as the cause of the disparity.

**Strengths And Weaknesses:**

Strengths:
1. This paper is well written and the idea is clear and easy to follow.
2. It is novel to explain the fairness with gradient norms and Hessian matrices across groups theoretically.
3. The proposed method is simple yet efficient to mitigate the disparity across groups.
Please look at the Questions for the weaknesses.

---

> ### Author Response · Authors · 2022-08-01
> **Authors' response**
>
> We thank the reviewer for their time and we appreciate your comments on our efforts in exploring the proposed idea of fairness in pruned models. We answer your questions below and will be happy to answer any further questions or doubts you may have.
>
> **Q1**
>
> The mitigation solution explored (e.g., solving Problem (8)) is applied either to the original large network (before pruning) or
> to the pruned network, after pruning. The former requires retraining the original model. The latter, instead, is more practical and only requires a retraining step on the pruned network.
>
> Note that *all* networks and methods illustrated in Figures 8 and 9 undergo a process of fine-tuning, which has been shown to improve the performance of the pruned networks. However, while _Fair  Bf  Pruning_ applies a standard fine-tuning step, _Fair  Aft  Pruning_ and _Fair  Both_ apply the constrained fine-tuning step illustrated in Problem (8).
>
> **Q2**
>
> Figure 7 is used to illustrate the effectiveness of the proposed solution: Despite implementing a relaxation (problem (8)) of the original constrained empirical risk problem (problem (7)), we are able to effectively reduce the inequalities among gradient norms (middle sub-plot) as well as those among group Hessian (right sub-plot).
>
> Both sub-figures (top and bottom) use an _unpruned_ model. The top sub-figure shows the performance of a classical unpruned model trained by minimizing the empirical risk function. The bottom sub-figure shows the effect of solving Problem (8), i.e., it constrains the empirical risk function with the various group loss terms.
> We believe this could have been conveyed more explicitly to the readers and will add a sentence to clarify it further in the final version. Thank you for pointing this out!
>
> **Q3**
>
> The reported experiments illustrate tasks in which the protected attributes are also the task labels. However, we have run additional experiments; Our extended results on using different protected attributes and labels show very similar trends to those reported in the paper. This speaks to the effectiveness of the proposed mitigation solution, and we will include them in the final version.
>
> Below we show the results for the UTKFaces dataset with `ethnicity` as class labels and `age` as protected attributes (exactly as you mentioned) for a CNN with two Convolutional layers and three linear layers and prune amounts: [30%, 50%, 70%, 90%]. This result aligns with our previous observations where `Fair Aft pruning` and `Fair both` achieve relatively lesser fairness violations compared to the `No mitigation` and the `Fair bf Pruning` methods in most cases.
>
> | Methods            | Overall Accuracy           | Fairness violations           |
> |--------------------|----------------------------|-------------------------------|
> | No mitigation      | 0.546, 0.545, 0.529, 0.559 | 0.179, 0.186, **0.152**, 0.134    |
> | Fair bf Pruning    | 0.539, 0.557, 0.529, 0.54  | 0.189, 0.190, 0.174, 0.238    |
> | Fair Aft Pruning | 0.538, 0.532, 0.497, 0.472 | 0.172, 0.161, 0.163, **0.05** |
> | Fair both          | 0.525, 0.541, 0.508, 0.484 | **0.170**, **0.144**, 0.156, 0.073    |
>
> On top of that, we also run similar experiments using CELEB-A dataset with the VGG-19 model; Again the trends are similar and we will also report these new experiments in the final version of the paper.
>
> **Q4**
>
> The labels reported under the components of Equation (6) are used to provide intuitive meaning about the semantics of those terms, not their exact formula. To avoid any confusion, in the final version, we will modify these labels as,
> -  "accuracy"  ->  "error"
> -  "distance  to  the  decision  boundary"  ->  "closeness  to  the  decision  boundary"
>
> **Q5**
>
> Soft output, in this paper, refers to the predictions *after* the softmax layer.
>
> We hope to have clarified your doubts comprehensively and are happy to discuss any further questions. In light of these clarifications, and on our observation that the reviewer reported no weaknesses in their review, we also hope they consider raising their score.

---

> > ### Comment · Reviewer_3Nrs · 2022-08-09
> > **Thanks for the clarification.**
> >
> > I would like to thank the authors for their clarifications. As the authors mentioned, I hope some of these discussions could be included in the camera-ready version of your paper.
> >
> > I have one additional question for Figure 7. What's the overall accuracy of the unfair model (top) and the fair counterpart (bottom)? It looks like the overall accuracy of the fair model decreased compared to the unfair model. If so, please further explain why the proposed loss may decrease the performance.

---

> > > ### Author Response · Authors · 2022-08-09
> > > **Response**
> > >
> > > We thank the reviewer for their comments and are happy our paper was well received.
> > >
> > > We will of course include our discussion in the final version of the paper.
> > >
> > > As we apply the constraints in equation (8) we enforce a process to minimize fairness violations of every group. This is a penalty that affects the total model loss function, and in turn, such loss will generally be greater than the loss function computed without enforcing a fairness constraint. This leads to a slight decrease in accuracy while preserving fairness. We will elaborate on this aspect in our final version with a brief statement.
> > >
> > > We hope that we have addressed your concern and we are happy to answer any further questions that you may have. Once again, we'd be glad if our clarification can be reflected in your final score on this paper.
> > >
> > > Many thanks!

---

> ### Author Response · Authors · 2022-08-07
> **Response**
>
> Dear reviewer,
>
> We hope we have clarified all your concerns. We know time is scarce and your comments are greatly appreciated. We also hope our clarifications could be reflected in the final score and will be happy to discuss any further comments you may have.
>
> Many thanks!

---

### Official Review · Reviewer_Zs7H · 2022-07-18

**Rating:** 7
**Confidence:** 4
**Ethics Flag:** Yes
**Soundness:** 2 fair
**Presentation:** 3 good
**Contribution:** 3 good

**Summary:**

The work studies effect of pruning neural networks on their losses for specific groups and the differences between them, termed as unfairness. It shows that gradient norm and Hessian of the loss function are important quantities controlling the upper bound on group-specific losses. As a way to mitigate unfairness from pruning, the effect of adding constraints to the learning problem that penalise the two quantities is explored. Some of the claims are validated through experiments on a face images used to train an ethnicity detection model.

**Questions:**

Major points to address in the response

1. There is an unacknowledged jump in reasoning from Corollary 1 to the main claim of the work that pruning increases unfairness. Corollary 1 shows that the upper bound on excessive loss for *a given* sensitive group may increase due to pruning. But it does not guarantee that *relative* excessive loss may increase. Also, the results are on the upper bound of the excessive loss and not the actual value of the excessive loss. The difference in actual loss across groups need not be smaller than the difference between upper bound. The theoretical results may motivate investigating gradient norms and the Hessian in experiments but the motivation is not as clear as presented. Thus the claims for studying disparate excessive loss in the rest of the results are overstated. The claims should be stated more carefully.

2. Please discuss the work by Hooker et al. 2020 which seems to have the same conclusion. Please discuss relationship between the findings.
Hooker et al. 2020, Characterising Bias in Compressed Models. https://arxiv.org/abs/2010.03058

Another related work is Ahia et al. 2021 which posits that pruning affects infrequent sentences more. This relates to Proposition 1’s claim that the proportion of data points from the group is a potential source of disparate impact of pruning.
Ahia et al. 2021, The Low-Resource Double Bind: An Empirical Study of Pruning for Low-Resource Machine Translation. https://aclanthology.org/2021.findings-emnlp.282/

3. Validating the relaxed version of the mitigation method eq. (8) is a bit unsatisfactory. Partly because it is not directly related to the analysis. Consider solving the optimization problem in eq. (7) instead of the relaxed version on a smaller dataset and model. This will help in validating the constraints motivated by the theoretical analysis instead of simplifying the problem. Perhaps only the gradient norm penalty, which is relatively quicker to estimate than max eigenvalue of the Hessian, can be tested in experiments.

Secondly, the relaxed version is an existing fairness method which is not discussed. It is also considered in the domain generalization problem where one approach is to equalize losses across domains that are groups in this case. In this sense, the contribution can be rephrased as taking an existing method and showing its relevance for fairness of pruning.
See Donini et al. 2018 https://papers.nips.cc/paper/2018/hash/83cdcec08fbf90370fcf53bdd56604ff-Abstract.html, and Williamson and Menon 2019 https://proceedings.mlr.press/v97/williamson19a.html for the fairness method, and Section 2.4 of Krueger et al. 2021 https://proceedings.mlr.press/v139/krueger21a.html for the domain generalization method.

4. Finally, please address the ethical concern of using the UTKFace dataset as the only exemplar in the work.


Minor points or suggestions (no response is being expected)

Handling one failure mode (disparity due to pruning, here) in isolation risks unintended consequences on other failure modes. Please consider adding a note that the question of how the proposed solution impacts fairness on other attributes, robustness, and interpretability of the pruned network.

Given the discrepancy between the theoretical results and claims, the empirical results become essential to the significance of the work. Consider repeating the analyses in Figures 2 to 6 on more models and datasets as considered in Figure 9.

In the line 88, fairness wrt pruning is taken to be zero excessive loss \xi(D) =0. While terming this as fair pruning is ok, it can be commented that the pruned network can still be unfair in terms of differences in losses between groups.

Clarify the accuracy metric used in Figure 3.

Mention the discrepancy between loss metrics (like squared loss) used in the results and accuracy metric (0-1 loss) used to evaluate them in the experiments.

Proposition 3 can be a remark as it is a known fact.

Add error bars specifically in Figure 8 and 9 where different methods are compared. Ideally error bars should be reported in other figures also to convey robustness of the findings.

I think a more natural way to visualise Figure 1 would be a multiple line plot of accuracy vs pruning percentage with separate lines for each group.

Report the R^2 value for the correlation between the two variables shown in Figure 2, 3 and 6 which is easier to interpret.

It makes more sense to normalize the gradient norm plot in Figure 8 since two different models are compared.

Include a reference to Weyl’s inequality e.g. ones in https://terrytao.wordpress.com/tag/weyl-inequalities/.
Weily’s -> Weyl’s in Appendix.

Consider using the first person in sentences as it is less ambiguous, The paper uses -> we use.

**Ethics Review Area:**

["Inappropriate Potential Applications & Impact  (e.g., human rights concerns)"]

**Limitations:**

Some of the limitations are presented. Please discuss limitations including relevance of theoretical results to differences in excessive loss, reliance on assumptions like being able to find the local minima. Also, mention the effect of other pruning methods, and uncertainty around effect on other properties of the pruned model than fairness. Prominently write the assumptions in one place and reference them in the results like in Proposition 1 which does not state the assumptions.

Please consider clarifying why ethnicity detection task from face images was used as an example to illustrate effects of pruning. Since it is a controversial application of machine learning with damaging consequences (with or without fairness concerns), it becomes imperative to motivate its inclusion in the paper. In my personal opinion (disagreements are welcome), the inclusion of facial recognition application should not appear as an endorsement of the technology.

**Strengths And Weaknesses:**

Strengths
- Provides clear experimental evidence that backs the claim of relation between excessive losses from pruning and properties of the network such as gradient norm and the max eigenvalue of the Hessian.
- Concepts and results are presented clearly.
- Studies a practically relevant problem.

Weaknesses
- The main claim on unfairness from pruning is not well-motivated by the theoretical results. The discussion centres on unfairness that is difference in excessive losses across groups while the result analyzes excessive losses for each group separately.
- Missed related work which makes a similar observation that pruning may cause unfairness.
- Experimental results are not presented as clearly as possible. Error bars can be reported and correlation coefficient can be reported in plots where the claim is that the plotted quantities are correlated.

The evidence presented in this work that pruning causes unfairness is an important one because of the prevalence of pruning in practice. Also, the observations on gradient norm and max eigenvalue of the Hessian are interesting. However, the presentation of the results seems to overstate the significance of the theoretical results. I think that these can be addressed through minor writing edits.

---

After the response

The response adequately clears my concerns on the proposed relaxation for the solution and related work. The response to the over-interpretation of the theoretical result in the paper is adequate. The promised change to the text after Corollary 1 is good. I would strongly encourage the authors to propagate this change to the rest of the paper e.g. the text in the heading of Sections 4 and 5 can be changed to may cause.

The work presents concrete evidence for the relation between two training properties during pruning and unfairness which is an important problem due to prevalence of pruning. Thus I am increasing the score to 7, Accept. The motivation from theory needs improvement which needs to be adequately acknowledged in a revision. The presentation should be improved also including error bars in figures.

---

> ### Author Response · Authors · 2022-08-01
> **Authors' response**
>
> Thank you for the constructive feedback. We are glad the paper was well-received. We, next, address your comments and questions and will be happy to discuss and clarify any further doubts.
>
> **Claims:**
>
> The reviewer correctly points out that the corollary deals with upper bounds on the excess loss. As noted in the paper, there is a relative change of excess risk due to pruning, which in turn, may exacerbate unfairness. While we are careful at analyzing and describing the causes for such unfairness, we also agree that some statements could be (and will be) further emphasized to clarify the dependencies between the various factors.
>
> **Related  work:**
>
> Thank you for pointing out this important aspect. We acknowledge this point and, in fact, we had already revised our paper (albeit after submission) with the following paragraph in the related work section:
>
> >  Network  compression  has  also  been  shown  to  have  a  profound  impact  towards  model  fairness.  For  example,  several  works  observed  empirically  that  network  compression  may  amplify  unfairness  in  different  learning  tasks  [1,  2,  3,  4].  Most  of  the  focus  has  been  on  vision  tasks  and  on  identifying  the  set  of  _Identified  Exemplars_  (IEs),  the  samples  that  are  mostly  impacted  under  the  compression  scheme.  These  works  find  that  IEs  belong  to  low-frequency  groups  (those  observed  at  the  tail  of  the  data  distribution)  [2,  3].  Blakeney  et  al.  [5]  further  investigate  how  bias  could  be  evaluated  and  mitigated  in  pruned  neural  networks  using  knowledge  distillation,  while  Hosseini  et  al.  [6]  observed  empirically  that  knowledge  distillation  processes  may  produce  unfair  student  models.  The  impact  of  network  compression  towards  fairness  has  also  been  assessed  in  natural  language  tasks.  For  example,  Du  et  al.  [7]  and  Xu  et  al.  [8]  empirically  measure  the  robustness  of  compressed  large  language  models,  while  Ahia  et  al.  [9]  look  into  how  compression  schemes  affect  data-limited  regimes.  Finally,  Xu  and  Hu  [10]  investigate  ways  to  improve  fairness  in  generative  language  models  by  compressing  them.
>
> We want to remark that, while important contributions to the state of knowledge, these works focus on providing empirical evidence for the unfairness arises. More precisely, these works experimentally observe that groups associated with low-frequency data are impacted more under model compression, albeit without providing a clear theoretical justification. We believe our contribution in this direction is unparalleled as it provides a clear link between the factors responsible for amplifying unfairness in model pruning. In particular, complementing the observations of the work pointed out by the reviewer, Proposition 1 illustrates how minority groups tend to have a larger gradient norm when compared with majority groups and are thus further penalized by the effect of pruning.
>
> Our work also goes beyond this observation and not only shows that both gradient norm and Hessian of the loss function are important quantities controlling the group-specific excessive loss, but also provides a mitigating solution to the unfairness effects. Crucially the design of this solution is justified by our theoretical findings.
>
> **Mitigating  solution**
>
> Indeed we have performed experiments on the “full” version of the mitigating solution (as detailed in Equation (7)) but reported those pertaining to the “relaxed” solution as computationally much faster and had noticed a negligible difference in results. We believe that the proposed solution is much more applicable in practice, and thus the reason why we have described it. For completeness and comparison, we will add the results of the full version in the appendix.
>
> **Ethical  statement**
>
> Thank you for pointing this out! All of the authors of this work are also strong privacy advocates. In fact, by no means should our results be intended to enforce the use of face recognition systems –- we will also explicitly mention this point to clarify further. Several vendors, unfortunately, have already developed and deployed such systems in practice. We hope our work creates further awareness of the unfairness caused by these systems under pruning and provides a deeper understanding of why these issues arise in the context of models that could be deployed in low-powered devices, such as smart cameras or access control systems.
>
> We agree with the ethical considerations of facial recognition and will add a solid statement to clarify our scope.
>
> **Minor  points**
>
> Thank you for the minor questions pointing out which will be addressed in depth.
>
> We hope to have clarified your doubts comprehensively and are happy to discuss any further questions. In light of these clarifications, we also hope the reviewer considers raising their score.

---

> > ### Author Response · Authors · 2022-08-04
> > **References**
> >
> > We report below the references associated with the `Related work` section in the previous response.
> >
> > **References**
> >
> > [1] - M. Paganini. "Prune responsibly." arXiv preprint arXiv:2009.09936 (2020).
> >
> > [2] - S. Hooker, A. C. Courville, G. Clark, Y. Dauphin, and A. Frome.  "What do compressed deep neural networks forget?." arXiv preprint arXiv:1911.05248 (2019).
> >
> > [3] - S. Hooker, N. Moorosi, G. Clark, S. Bengio, and E. L. Denton. "Characterising bias in compressed models". arXiv preprint arXiv:2010.03058 (2020).
> >
> > [4] - V. Joseph, S. A. Siddiqui, A. Bhaskara, G. Gopalakrishnan, S. Muralidharan, M. Garland, S. Ahmed, and A. R. Dengel. "Going beyond classification accuracy metrics in model compression". arXiv preprint arXiv:2012.01604 (2020).
> >
> > [5] - C. Blakeney, N. Huish, Y. Yan, and Z. Zong. "Simon says: Evaluating and mitigating bias in pruned neural networks with knowledge distillation". arXiv preprint, arXiv:2106.07849 (2021).
> >
> > [6] - S. Hosseini, M. A. Shabani, M. M. Jahanara, and B. Salamatian. "Learning fair from unfair teachers". (http://www.sfu.ca/~mjahanar/learning_fair_from_unfair_teacher.pdf)
> >
> > [7] - M. Du, S. Mukherjee, Y. Cheng, M. Shokouhi, X. Hu, and A. H. Awadallah. "What do compressed large language models forget? robustness challenges in model compression". arXiv preprint arXiv:2110.08419 (2021).
> >
> > [8] - H. Xu, X. Liu, Y. Li, A. K. Jain, and J. Tang. "To be robust or to be fair: Towards fairness in adversarial training", Proceedings of the 38th International Conference on Machine Learning, PMLR 139:11492-11501, (2021).
> >
> > [9] - O. Ahia, J. Kreutzer, and S. Hooker. "The low-resource double bind: An empirical study of pruning for low-resource machine translation". In Conference on Empirical Methods in Natural Language Processing (EMNLP), 2021.
> >
> > [10] - G. Xu and Q. Hu. "Can model compression improve nlp fairness." arXiv preprint arXiv:2201.08542 (2022).
> >
> > Again, we thank the reviewer for their time and effort and are happy to discuss any further questions.

---

> ### Author Response · Authors · 2022-08-07
> **Response**
>
> Dear reviewer,
>
> We hope we have clarified all your concerns. We know time is scarce and your comments are greatly appreciated. We also hope our clarifications could be reflected in the final score and will be happy to discuss any further comments you may have.
>
> Many thanks!

---

> ### Comment · Reviewer_Zs7H · 2022-08-07
> **On the response**
>
> Thank you for responding to the queries. The following two queries are not addressed. The ones on related work are addressed perfectly.
>
> ### Claims
>
> I think the response missed to address my point. My concern is that Corollary 1 analyzes excess risk for a group but the claim of increase in unfairness is on the difference between excess risks between groups. That is, the excess risk for both groups may increase after pruning while leaving the difference between them unchanged. So the unfairness may or may not increase. I am ok with the analysis proving gradient and Hessian properties as causes for the increase in excess risk for a group (it is a good contribution). But the discussion in the paper incorrectly implies that the unfairness increases and it is due to the two properties. So the theoretical results are over-interpreted. In light of this discrepancy, the 'may exacerbate' in this statement needs to be more concretely specified.
> > As noted in the paper, there is a relative change of excess risk due to pruning, which in turn, may exacerbate unfairness.
>
> ### Mitigating solution
>
> Comparison between "full" and "relaxed" version could have been included in the response either in the response or in the updated PDF. For example, percentage changes in performance metrics when switching from relaxing one or both of the constraints would help. The relation between the relaxed version and the existing fairness and domain generalization approaches will put the method in a broader context. It does not decrease the contribution of this work.

---

> > ### Author Response · Authors · 2022-08-09
> > **Response**
> >
> >
> > We thank the reviewer for their time and response and provide below
> > further clarification on the questions raised.
> >
> > ## Claims
> >
> > We agree with the point raised by the reviewer on that the explanation
> > sentence right after Corollary 1 may be misleading and will revise it
> > as follows:
> >
> > > The corollary above indicates that the excess risk for a group
> > > increases as the pruning regime increase. While this result pertains
> > > to the loss in accuracy due to pruning for a group, the paper
> > > illustrates next why unfairness can become more significant as the
> > > pruning regime increases.
> >
> > As noted in lines 171--178, note that Theorem 1 and Corollary 1
> > show that the excessive loss of a group is controlled by term
> > $\|\mathbf{g}^\ell_a\| \times \| \bar{\theta} - \theta^*\|$.
> >
> > Notice that groups with different gradient norms will have a
> > disparate effect on the resulting term above (and thus on their
> > excessive loss). In particular, groups with very small gradients norms (those
> > generally associated with highly accurate predictions) will be less
> > sensitive to the effects of the pruning rate (regulated by term
> > $\|\mathbf{g}^\ell_a\|$. Conversely, groups with large gradient norms
> > will be affected by the pruning rate to a greater extent, with larger
> > pruning rates, typically reflected in larger excessive losses.
> >
> > A similar analysis is made for the Hessian loss term in lines 238--249.
> >
> > Finally, we also note that the upper bound provided in Theorem 1 is tight
> > for linear regression with least square loss. Under this setting, the
> > third derivative term in Theorem 1 disappears. As a consequence, in this
> > setting, the unfairness might increase according to the pruning rate
> > when the difference in gradients/Hessian components across groups is
> > significant. In fact, we note that pruning in the context of linear models
> > is equivalent to performing a feature selection step.
> >
> > ## Full vs relaxed mitigation
> >
> > The following table compares the accuracy of the "full"-version with the
> > "relaxed" mitigating solution.
> >
> > | Datasets      | Class-wise accuracy in "full" version                                | Class-wise accuracy in "relaxed" version                             | Overall Accuracy ("full" version) | Overall Accuracy ("relaxed" version) |
> > |---------------|----------------------------------------------------------------------|----------------------------------------------------------------------|-----------------------------------|--------------------------------------|
> > | UTK-age_bins  | 0.856, 0.128, 0.145, 0.319, 0.331, 0.342, 0.181, 0.334, 0.512        | 0.810, 0.096, 0.141, 0.284, 0.385, 0.324, 0.227, 0.257, 0.533        | 0.395                             | 0.390                                |
> > | UTK-gender    | 0.830, 0.876                                                         | 0.868, 0.845                                                         | 0.857                             | 0.852                                |
> > | SVHN          | 0.864, 0.911, 0.869, 0.819, 0.887, 0.784, 0.840, 0.877, 0.805, 0.856 | 0.824, 0.91, 0.775, 0.726, 0.827, 0.752, 0.747, 0.789, 0.713, 0.755  | 0.857                             | 0.795                                |
> > | MNIST         | 0.998, 0.996, 0.993, 0.998, 0.994, 0.991, 0.991, 0.993, 0.992, 0.985 | 0.994, 0.988, 0.989, 0.986, 0.987, 0.979, 0.981, 0.988, 0.969, 0.994 | 0.993                             | 0.986                                |
> >
> > We note that while the "full" version leads to fairer results, the reduction in the
> > various groups accuracy is often insubstantial.
> > We also note that the running time of the "full" version is largely (over an order magnitude)
> > longer than the relaxed counterpart. This is due to the calculation of gradient norms
> > and the Hessian terms associated with each group.
> > Again, we displayed the "relaxed" version only in the main paper as we believed
> > more practical while at a contained decrease in accuracy.
> >
> > The final paper will report not only the table above but also a more comprehensive
> > version of these results which exclude only ONE constraint in the full version.
> > We are currently running these experiments.
> >
> >
> > We hope to have provided clarifications to your doubts and are happy to
> > discuss any further questions.
> >
> > Many thanks!

---

> > > ### Comment · Reviewer_Zs7H · 2022-08-09
> > > **Thanks for the additional response**
> > >
> > > The response adequately clears my concerns on the proposed relaxation for the solution and related work. Thanks for including the comparison table. The response to the over-interpretation of the theoretical result in the paper is adequate. The promised change to the text after Corollary 1 is good. I would strongly encourage the authors to propagate this change to the rest of the paper e.g. the text in the heading of Sections 4 and 5 can be changed to may cause.
> > >
> > > The work presents concrete evidence for the relation between two training properties during pruning and unfairness which is an important problem due to prevalence of pruning. Thus I am increasing the score to 7, Accept. The motivation from theory needs improvement which needs to be adequately acknowledged in a revision. The presentation should be improved also including error bars in figures.

---

> > > > ### Author Response · Authors · 2022-08-09
> > > > **Response**
> > > >
> > > > Many thanks again for the thorough and constructive review and discussion.
> > > >
> > > > We will of course address these changes in-depth in the final version of the paper and also add error bars to the figures.
> > > >
> > > > Thank you!

---

### Review · Ethics_Reviewer_LLGy · 2022-07-31

**Recommendation:**

The authors may consider swapping out the example task for another one. They should certainly discuss in a few sentences the potential issues with the task they are illustrating the method on (in the broader impacts section).

**Ethical Issues:**

Yes

**Ethics Review:**

Although the paper is a study of bias mitigation, as one reviewer points out, the ethnicity classification task from face images is already flawed in some ways dealing with surveillance and enabling oppression. (Even the much-lauded Gender Shades paper has a problematic task underlying it.)

---

### Review · Ethics_Reviewer_PMUh · 2022-08-13

**Recommendation:**

I'm not sure that it will be possible with the remaining time that the authors have.

**Ethical Issues:**

Yes

**Ethics Review:**

My guess is that ethics review was requested for this paper on the basis of using face recognition datasets to train an ethnicity classifier - this is honestly a pretty risky decision on the part of the authors and I would recommend not doing this.
Although the point the authors are making is good, that the pruned models are by default discriminatory but can be less so using their method, the choice of task is pretty risky and IMHO not the best way to show the issue.
I would propose going one of two ways:
1) Produce synthetic data that follows some distributions that the authors are finding from real-world datasets (which can be ethnicity) and assign each of them a label.
2) Use a different task that does not necessarily deal with humans or at least deals with it in a different way. For instance the CelebA dataset contains a column for whether the person is wearing a hat or not.

The larger point that discrimination is exacerbated through pruning and can be addressed, can be written without using such a risk-filled task. In fact, a single paragraph example can be given of face and ethnicity recognition (with the ethical consideration section the authors are proposing), to show the issue also applies to humans in general. But I think centring that creates unnecessary risks.

---

### Meta-Review · Area_Chair_zdrN · 2022-08-28

**Recommendation:** Accept
**Confidence:** Certain

**Metareview:**

The paper hits two important angles in current ML: first, offering theoretical predictions about the behavior of networks in the context of pruning, and second, providing guidance on how differences in data distributions will be affected by pruning. Examples include face image analysis with labels for race, these should be treated carefully so as not to pass on errors.

Ethics reviewers would prefer that face recognition not be the central example, but serve more as a supporting example. If that modification is possible within the NeurIPS timeframe if the paper is accepted, I would like to see that happen, but I would not necessarily hold back on publication for that sole reason. At the very minimum there should be a discussion of potential harms of disparate quality and availability in facial recognition technology, and how the current work relates to those. For example, do we *want* facial recognition to work well? Can we spell out the possible harms of disparate quality, such as increased risk of false identification?

The reviewers have engaged with author comments and have specific recommendations for improvements in clarity and evaluation strength.

**Award:**

Yes

---

### Decision · Program_Chairs · 2022-09-14

Accept